# Free-standing ultrathin silicon wafers and solar cells through edges reinforcement

Taojian Wu[1,5], Zhaolang Liu[2,5], Hao Lin [2,3] ✉, Pingqi Gao [2,3,4] ✉ & Wenzhong Shen [1] ✉

Crystalline silicon solar cells with regular rigidity characteristics dominate the photovoltaic market, while lightweight and flexible thin crystalline silicon solar cells with significant market potential have not yet been widely developed. This is mainly caused by the brittleness of silicon wafers and the lack of a solution that can well address the high breakage rate during thin solar cells fabrication. Here, we present a thin silicon with reinforced ring (TSRR) structure, which is successfully used to prepare free-standing 4.7-μm 4-inch silicon wafers. Experiments and simulations of mechanical properties for both TSRR and conventional thin silicon structures confirm the supporting role of reinforced ring, which can share stress throughout the solar cell preparation and thus suppressing breakage rate. Furthermore, with the help of TSRR structure, an efficiency of 20.33% (certified 20.05%) is achieved on 28-μm silicon solar cell with a breakage rate of ~0%. Combining the simulations of optoelectrical properties for TSRR solar cell, the results indicate high efficiency can be realized by TSRR structure with a suitable width of the ring. Finally, we prepare 50 ~ 60-μm textured 182 × 182 mm² TSRR wafers and perform key manufacturing processes, confirming the industrial compatibility of the TSRR method.

Photovoltaics plays a leading role in achieving the goal of a low-carbon-emission society. Nowadays, crystalline silicon (c-Si) solar cell dominates the photovoltaic (PV) market, with a market share of over 95% owing to their high module efficiencies, long lifespan of more than 25 years as well as high abundance of silicon[1]. Among them, there is a huge market potential for lightweight and flexible thin c-Si solar cells since they can be integrated with buildings, remote power applications such as electric vehicles and aircrafts[2] and wearable electronic devices[3]. However, they are not yet widely used due to the mechanically brittle nature of c-Si[4] and the dramatically increasing trend in breakage rate during cell processing as the thickness of the wafer decreases[5,6].

For the above reason, there is a trade-off between thickness and area for thin silicon solar cells. It is very challenging to prepare thin c-Si solar cells with large areas to a very thin thickness. Table 1 summarizes the characteristics of c-Si solar cells with a thickness of ≤ 40 μm reported since 2010. We can see that the vast proportion of the solar cells has an active area of less than 4 cm², and some of them even have an area of less than 0.03 cm². At present, the most straight-forward and low-cost route for preparing thin silicon (solar cells) is to process them in a free-standing way, as is done for standard wafers. However, in 2016, CEA-INES reported a drastic increase in the breakage rate from about 10% at 100 μm thickness to a terrible ~96% at 70 μm thickness in their silicon heterojunction (SHJ) pilot line (using 156 × 156 mm²

[1]Institute of Solar Energy, Key Laboratory of Artificial Structures and Quantum Control (Ministry of Education), School of Physics and Astronomy, Shanghai Jiao Tong University, 800 Dong Chuan Road, Shanghai 200240, China. [2]School of Materials, Shenzhen Campus of Sun Yat-sen University, No. 66, Gongchang Road, Shenzhen, Guangdong 518107, China. [3]Institute for Solar Energy Systems, State Key Laboratory of Optoelectronic Materials and Technologies, Sun Yat-sen University, Guangzhou 510275, China. [4]Jiangsu Collaborative Innovation Center of Photovoltaic Science and Engineering, Changzhou University, Changzhou 213164, China. [5]These authors contributed equally: Taojian Wu, Zhaolang Liu. ✉e-mail: linh229@mail.sysu.edu.cn; gaopq3@mail.sysu.edu.cn; wzshen@sjtu.edu.cn

**Table 1 | Characteristics of crystalline silicon solar cells with a thickness of ≤ 40 µm reported since 2010**

| Thickness (µm) | Methods for thin silicon | Active area (cm²) | Silicon technology | $J_{SC}$ (mA·cm⁻²) | $V_{OC}$ (mV) | FF (%) | η (%) | Remarks |
|---|---|---|---|---|---|---|---|---|
| 40 | Alkali etching | 244.3 | SHJ | 34.7 | 729.8 | 73.3 | 18.6 | Free-standing; Ref. 45. |
| 37 | Alkali etching | 4 | SHJ | 33.2 | 697 | 65.5 | 15.1 | Free-standing; Ref. 19. |
| 35 | Epitaxy | 239.7 | SHJ | 38.5 | 687 | 80.3 | 21.2 | Ref. 40. |
| 30 | Epitaxy | 70 | HJ | 31.7 | 634 | 80.8 | 16.2 | Ref. 46. |
| 30 | Alkali etching | N/A | HJ | 31.4 | 495 | 75.4 | 11.7 | Free-standing; Flexible; Taped on PET during processing; Ref. 47. |
| 25 | SOM | 1.1 | SHJ | 33.6 | 580 | 76.7 | 14.9 | Ref. 48. |
| 25 | Alkali etching | 1 | DF | 31.9 | 626 | 75.6 | 15.1 | Free-standing; Ref. 49. |
| 22.5 | Alkali etching | 4 | HJ + DF | 33.2 | 555 | 76 | 14.0 | Ref. 50. |
| 20 | Alkali etching | 1 | SHJ | 30.3 | 699 | 77.1 | 16.3 | Free-standing; Flexible; Ref. 18. |
| 20 | Alkali etching | N/A | DF | 32.1 | 564 | 75.2 | 13.6 | Free-standing; Ref. 23. |
| 20 | N/A | N/A | HJ | 32.5 | 629 | 73.9 | 15.1 | Free-standing; Ref. 20. |
| 18 | Epitaxy | 4 | HJ | 34.5 | 632 | 77.2 | 16.8 | Ref. 51. |
| 15 | Micro-machining | 0.00045 | HJ | 23.0-26.0 | 440 ~ 480 | 67.0 ~ 68.0 | 6.0 ~ 8.0 | Transfer printing onto PDMS substrate; Flexible; Mini-module; Ref. 52. |
| 15 | Micro-machining | 1 | HJ | 6.5 | 504 | 51.5 | 1.7 | Encapsulated by PDMS; Flexible; Ref. 53. |
| 14.8 | Alkali etching | 0.23 | DF | 25.8 | 550 | 46.6 | 6.6 | Free-standing; Flexible; Ref. 54. |
| 14 | Five etches | 0.0004 | HJ + IBC | 31.8 | 597 | 78.4 | 14.9 | Ref. 55. |
| 14 | Alkali etching | 0.8 | DF | 21.3 | 560 | 76 | 9.1 | Free-standing; Flexible; Ref. 21. |
| 10 | SOI | 1 | HJ | 33.9 | 589 | 78.5 | 15.7 | Ref. 11. |
| 10 | SOI | N/A | HJ + IBC | 29 | 623 | 76 | 13.7 | Ref. 13. |
| 8.6 | Alkali etching | N/A | DF | N/A | N/A | N/A | 6.6 | Free-standing; Flexible; Taped on dummy wafer during processing; Ref. 22. |
| 8 | SOI | N/A | HJ | 16.5 | 525 | 55.9 | 4.8 | Ref. 17. |
| 8 | Micro-machining | 0.0005 | HJ | 40.1 | 473 | 65.4 | 12.4 | Transfer printing onto PET substrate; Flexible; mini-module; Ref. 24. |
| 6.8 | Alkali etching | N/A | HJ | 19.1 | 559 | 58 | 6.2 | Free-standing; Flexible; Taped on thick wafer during processing; Ref. 10. |
| 6.8 | Alkali etching | 0.23 | DF | 24.19 | 427 | 41 | 4.2 | Free-standing; Flexible; Ref. 54. |
| 5 | SOI | N/A | HJ | 26.4 | 590 | 69 | 10.8 | Ref. 16. |
| 3.7 | Alkali etching | N/A | HJ | 12.9 | 474 | 74 | 4.5 | Free-standing; Flexible; Taped on thick wafer during processing; Ref. 10. |
| 3 | Micro-machining | 0.00015 | HJ | 24.6 | 494 | 71.5 | 8.5 | Transfer printing onto PET substrate; Flexible; mini-module; Ref. 14. |
| 3 | Epitaxy | 0.008 | SHJ | 18.3 | 490 | 68 | 6.1 | Ref. 15. |
| 3 | Spalling | 1 | N/A | 12.6 | 553 | 61.7 | 4.3 | Layer transfer; Flexible; Ref. 56. |
| 2.7 | Epitaxy | 0.0221 | HJ | 23.7 | 630 | 82.1 | 12.3 | Released from parent substrate; Free-standing; Flexible; Ref. 9. |
| 2.4 | Epitaxy | 4 | SHJ | 16.6 | 546 | 77 | 7.0 | Ref. 57. |
| 2 | Epitaxy | 0.0221 | HJ + DF | 21 | 625 | 81.9 | 10.8 | Ref. 58. |
| 1.7 | Epitaxy | 4 | SHJ | 16.1 | 501 | 78.6 | 6.4 | Ref. 57. |
| 1.1 | 'Epifree' | 1 | SHJ | 19.7 | 560 | 78.2 | 8.6 | Ref. 12. |
| 0.9 | Epitaxy | 4 | SHJ | 15 | 478 | 66 | 4.7 | Ref. 57. |
| 0.75 | Epitaxy | N/A | SHJ | 10.2 | 557 | 60 | 3.4 | Ref. 59. |

$J_{SC}$ Short-circuit current density, $V_{OC}$ Open-circuit voltage, FF Fill factor, η Efficiency, SHJ Silicon heterojunction, HJ Homojunction, DF Dopant-free, IBC Interdigitated back contacts, SOM Semiconductor-on-metal, SOI Silicon-on-insulator, PET Polyethylene terephthalate, PDMS Polydimethylsiloxane.

pseudo-square wafers)[6]. Even though process and handling adjustments were implemented, a breakage rate of up to 4.5% during the fabrication of 100 µm thick high-temperature diffused junction cells in the pilot production line of Hanwha Q CELLS was reported[7]. Such high breakage rates lead to unacceptable yield losses and high total production costs. A solution to this fragility is to prepare thin silicon based on a parent substrate, such as epitaxy, spalling, 'epifree', silicon-on-insulator (SOI, 'smart-cut' process) and micro-machining (Table 1), and some of them require further bonding or transferring of the thin silicon to external supporters to cope with the following solar cell preparation process. Nevertheless, these fabrication processes are too complicated, leading to questions about the viability of these processes for fabricating thin silicon solar cells in a cost-effective way at an industrial scale. Moreover, the thin silicon with hard substrate is not flexible, which limits its range of applications. The challenges of free-standing and supported processing of thin silicon remain to be answered[8].

The vast majority of reports are concerned with solving the problem of reduced light absorption in thin silicon solar cells[9–24], while very few works are devoted to addressing the problem of high

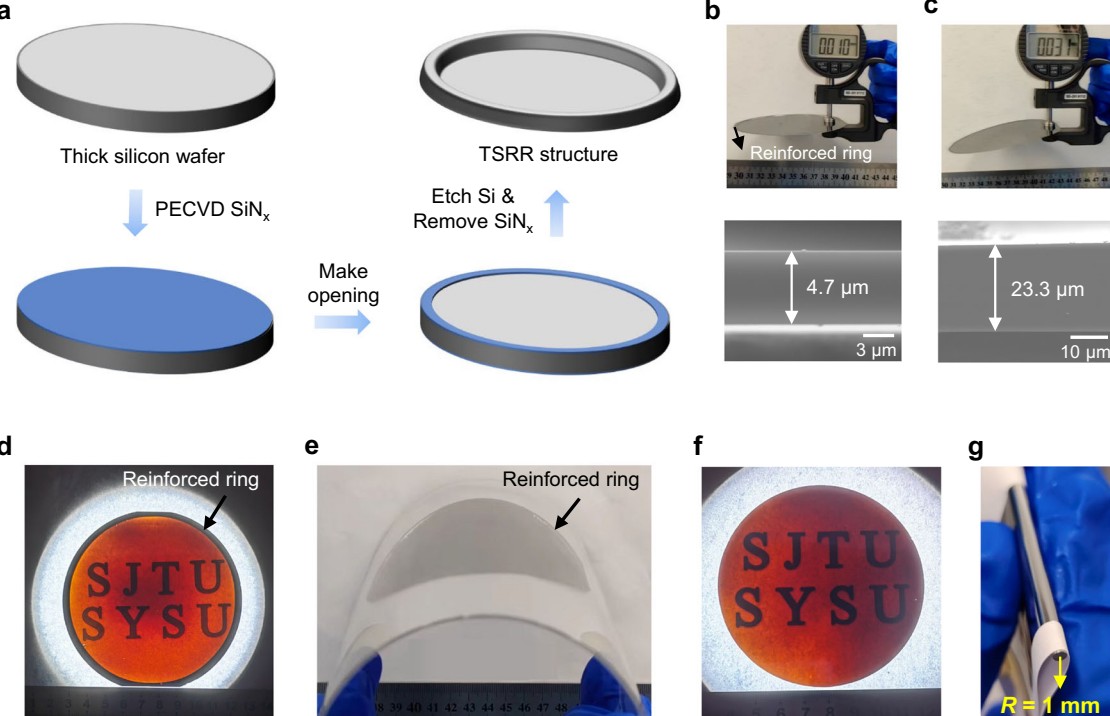

**Fig. 1 | Preparation and performance demonstration of the thin silicon with reinforced ring (TSRR) structure. a** Preparation process for TSRR structure. Thinned 4-inch wafers with **b** TSRR structure and **c** all-thin silicon (ATS) structure are measured by handheld thickness gauge (top) and the corresponding SEM images of the cross section (bottom). It should be noted that thinned 4-inch wafers with a thickness of <10 μm with ATS structure cannot be prepared using alkaline solution etching. **d** Optical image of the 4.7-μm 4-inch wafer with TSRR structure underneath white light illumination, the letters "SJTU, SYSU" on the paper below the thin silicon wafer are clearly visible. **e** Flexibility performance of the thinned wafer in **d**. Note that we can control the flexibility of the wafer with TSRR structure by adjusting the thickness of the reinforced ring. **f** Fully thin silicon wafer with an area of 60.8 cm² with the reinforced ring cut off and **g** its flexibility performance, the bending radius is 1 mm.

breakage rate during thin solar cell fabrication. For example, a locally thinned waffle-like cell was proposed for space silicon solar cell in 2000. Strobl et al. reported a 15.8% efficiency silicon solar cell with a thickness of 50 μm in the locally thinned regions and 130 μm for the frames[25]. But other details of this structure are particularly underreported. There is also a "3-D" wafer technology developed by 1366 technology, Inc. around 2016. It is a multi-crystalline silicon wafer growing technology which forms a wafer directly from molten silicon in a bath-like furnace, with the ability to locally control wafer thickness. Thus, it can produce thin wafers with thick edge[26,27]. However, it suffers from serious problems: low bulk lifetime, high total thickness variation (TTV) and difficulty in growing very thin framed wafers[27]. Recently, a technique of blunting pyramidal structure in the marginal regions was proposed by Liu et al. for thin silicon solar cells with a thickness of around 60 μm[2]. However, for thinner silicon wafers, there could be a lot of breakage before blunting pyramids.

In this contribution, we present a thin silicon with reinforced ring (TSRR) structure at the edge region, which can be used to prepare ultrathin silicon wafers with a large area and provide support throughout the solar cell preparation process to reduce the breakage rate. Then with the help of COMSOL Multiphysics, we investigated the mechanical properties of TSRR structure and the conventional all-thin silicon (ATS) structure, and the simulation results showed that the reinforced ring of TSRR structure can distribute a large amount of stress when subjected to external forces, thus making the central thin silicon region of TSRR structure bear a smaller force compared to ATS structure. We further prepared solar cells with TSRR structure, where all process steps are done in a free-standing way, and achieved an efficiency of 20.33% (certified 20.05%) on 28-μm silicon solar cell with all dopant-free and interdigitated back contacts. Meanwhile, the

breakage rate of each process step of solar cell fabrication with both structures was tracked. To gain an in-depth understanding of the effect of TSRR structure on the optoelectrical performance of solar cells, based on TCAD numerical simulations, we investigated the carrier transport mechanism of the solar cell with TSRR structure, and the impact of the thickness of the central thin silicon region and the width of the reinforced ring on the solar cell performance. Finally, we prepared 50 - 60-μm textured 182 × 182 mm² TSRR wafers and performed screen printing, high-temperature and wet manufacturing processes, which confirms the industrial compatibility of the TSRR method.

## Results

### Preparation of TSRR structure

Figure 1a shows preparation process for the TSRR structure. Generally, thick silicon wafers are etched into ATS wafers with desired thickness by alkaline solutions such as Potassium hydroxide (KOH) and Tetramethylammonium hydroxide (TMAH). However, the ATS structure is easily broken down during thin silicon solar cell fabrication, and it is important to note that it is not possible to prepare thinned 4-inch wafers with a thickness of <10 μm with ATS structure based on our experiments. We proposed a method to fabricate the TSRR structure, which requires only 3 more steps with common devices in photovoltaic factories for mass production: first depositing 70 nm silicon nitride (SiNx) on both sides of the normal thick silicon wafers by plasma enhanced chemical vapor deposition (PECVD) or low pressure chemical vapor deposition (LPCVD), then removing the SiNx from the central region of one side using a die, laser or photolithographic to make opening, and finally etching the wafer in alkaline solution to the desired thickness. Thanks to the protection of SiNx layer, the silicon in the edge region of the wafer maintains its original thickness, thus

forming a reinforced ring. Figure 1b (top) displays a thinned 4-inch wafer with TSRR structure being measured by a handheld thickness gauge (reading 10 μm), its real thickness is 4.7 μm, as depicted in Fig. 1b (bottom), which is the scanning electron microscope (SEM) image of the cross section. The width and thickness of the reinforced ring are 2 ~ 4 mm and 192 μm, respectively. To the best of our knowledge, this is the largest area of free-standing monocrystalline silicon with a thickness of <5 μm reported so far. Figure 1c (top) displays the corresponding measurement (reading 31 μm) by handheld thickness gauge of the thinned 4-inch wafer with ATS structure, and its exact thickness is 23.3 μm as shown in Fig. 1c (bottom). We can find that the thinned 23.3-μm wafer with ATS structure is bending downward under gravity only. However, this flexibility is not desired in the processing or testing stages, as it tends to cause breakage. In contrast, the 4.7-μm thin silicon wafer with reinforced ring still remains horizontal under gravity, demonstrating the supporting role of the reinforced ring. We also offer the SEM image about the boundary of the reinforced ring and the central thin silicon region in Supplementary Fig. 1.

Figure 1d is the optical image of the 4.7-μm 4-inch wafer with TSRR structure underneath white light illumination, the letters "SJTU, SYSU" on the paper below the thin silicon wafer are clearly visible, which reflects its ultra-thinness and high red light transmission[10]. At the same time, based on the color uniformity shown here, we can also see that the thickness of the wafer is fairly uniform, and according to our further quantitative measurements on the thickness uniformity, as demonstrated in Supplementary Fig. 2, the TTV for this thin silicon preparation method is within 6 μm. Despite of the thick reinforced ring, it is still bendable, as exhibited in Fig. 1e. And its flexibility depends on the combined bending performance of the reinforced ring and the central thin silicon region. This means that we can control the flexibility of the entire wafer by adjusting the thickness of the reinforced ring. If high bending performance is required, we can cut off the reinforced ring with a laser in the last process step. Figure 1f shows a fully thin wafer with an area of 60.8 cm$^2$ obtained in this way. And it has a bending radius of 1 mm as revealed in Fig. 1g.

## Mechanical properties of ATS and TSRR structures

Stress profile and deformation of ATS and TSRR structure in three cases during fabrication process in which the breakage rates are very high, were investigated by shell module under structural mechanics branch in COMSOL Multiphysics. Noted that since it is not possible to prepare thinned 4-inch wafers with a thickness of <10 μm with ATS structure as mentioned earlier, 30 μm is simulated for the ATS structure here. And the corresponding thickness is 30 μm for the central thin silicon region and 210 μm for the reinforced ring of the TSRR structure. All silicon wafers are 4 inches (10 cm) in size and the width of reinforced ring is 3 mm.

The first case is self-weight (handling or transferring). Figure 2a shows simplified schematic diagram of thin silicon wafer with a fixed position under the effect of gravity (corresponding to Fig. 1b and c). In the simulation, the vertical downward displacement of the point furthest from the fixed position is 14.3 mm for ATS structure and 2.5 mm for TSRR structure, as depicted in Supplementary Fig. 3. This is consistent with the experiments in which the displacement is ~16 mm for 24-μm wafer with ATS structure and ~3 mm for 6-μm wafer with TSRR structure. This confirms the validity of our simulation results. The Von Mises stress profile of the ATS structure (top) and the TSRR structure (bottom) in this case are demonstrated in Fig. 2b. The maximum stress is 7.21 × 10$^7$ N/m$^2$ = 72.1 MPa for ATS structure and the maximum stress point is located at the left end of the fixed position, i.e., near the overhanging area. In contrast, the maximum stress of TSRR structure is 51.8 MPa, which is smaller than 72.1 MPa for ATS structure, and maximum stress point is located at the right end of the fixed position, i.e., near the reinforced ring. In addition, the further away from the fixed position the less stress is suffered for ATS

structure, while this is not the case for TSRR structure. We can observe that the boundary between the reinforced ring and the central thin silicon region undergo a large amount of stress, which helps the central thin region to bear less stress than the ATS structure. A clearer comparison can be found in Fig. 2c, it is the stress distribution along the cut line σ$_1$ in Fig. 2b. The gray dashed line indicates the boundary between the reinforced ring and the central thin silicon region. It is clear that the stress in the central thin silicon region except near the boundary of TSRR structure are smaller than that of ATS structure. This fully illustrates the stress-sharing role of the reinforced ring.

Note that the state in Fig. 2a is similar to that when handling or transferring thin silicon wafers in experiments as the only external force during these processes is gravity. Handling and transferring have been experimentally demonstrated to be particularly prone to cause breakage for ATS structure[7,8], which means the breakage rate can be reduced during handling and transferring by using TSRR structure based on the results of Fig. 2b. In fact, as displayed in Supplementary Fig. 4, tweezers can be gripped at the reinforced ring instead of the central thin silicon region when we are handling or transferring the TSRR structure, which can further reduce the breakage rate.

The second case is wet process. In the preparation process of silicon solar cells, wet process is a necessary, such as anisotropic etching of silicon to form random pyramids, standard Radio Corporation of America (RCA) cleaning and simplest cleaning with deionized water. Figure 2d displays the states of thin silicon wafers during wet processing. Because the thin wafers are so light in mass, they float up due to buoyant force and bubbles (①). It was found through our experiments that a severe case is stiction due to surface tension, as this often leads to wafer breakage for ultrathin wafers when the edges of the wafers are constrained by the Teflon basket (②) or when trying to separate them. Assuming that the central circular area with a radius of 5 mm is subjected to a total force of 0.2 N along the direction perpendicular to the surface of the wafer during wet process. The Von Mises stress profile of the ATS (top) and the TSRR structure (bottom) in this case are demonstrated in Fig. 2e (note the outermost edge of the wafers is fixed). The maximum and minimum stresses are 316.0 MPa and 62.4 MPa respectively for ATS structure and the maximum stress point is located at the center of the wafer. With regard to TSRR structure, the maximum stress is only 10.2 MPa, which is down to 3% of the maximum stress of ATS structure, and this is even 83.7% smaller than the minimum stress of ATS structure. The maximum stress point is slightly deviated from the center of the wafer and closer to the reinforced ring. The minimum stress is 0.4 MPa, which is 2 orders of magnitude smaller than the minimum stress of ATS structure. Figure 2f provides a comparison of the stress distribution along the cut line σ$_2$ in Fig. 2e for the two structures. Obviously, the suffered stress of TSRR structure is always less than that of ATS structure. Similar to Fig. 2c, the stress increases sharply at the boundary of TSRR structure. In fact, based on our bending experiments of thin silicon with different thicknesses and corresponding simulation results offered in Supplementary Fig. 5, we obtain that the wafer breaks when the maximum Von Mises stress is greater than 345 ~ 533 MPa, which is compatible with the reported fracture strength of crystalline silicon with a value of 80 ~ 520 MPa (varying with the surface damage of silicon wafer)[28–31]. This means that the ATS structure with this state of stress shown in Fig. 2e (top) is likely broken, while the TSRR structure stays safe.

It is due to the stress-sharing effect of the reinforced ring that we can prepare 4.7-μm 4-inch wafers with TSRR structure, while it is not possible to prepare thinned 4-inch wafers with a thickness of <10 μm with ATS structure by alkaline solution etching wet process. We can clearly see the behaviors of both structures under complex stresses during the alkaline solution etching wet process in the Supplementary Movie 1.

The third case is screen printing. In the industrial production of large-area silicon solar cells, electrodes are prepared by screen

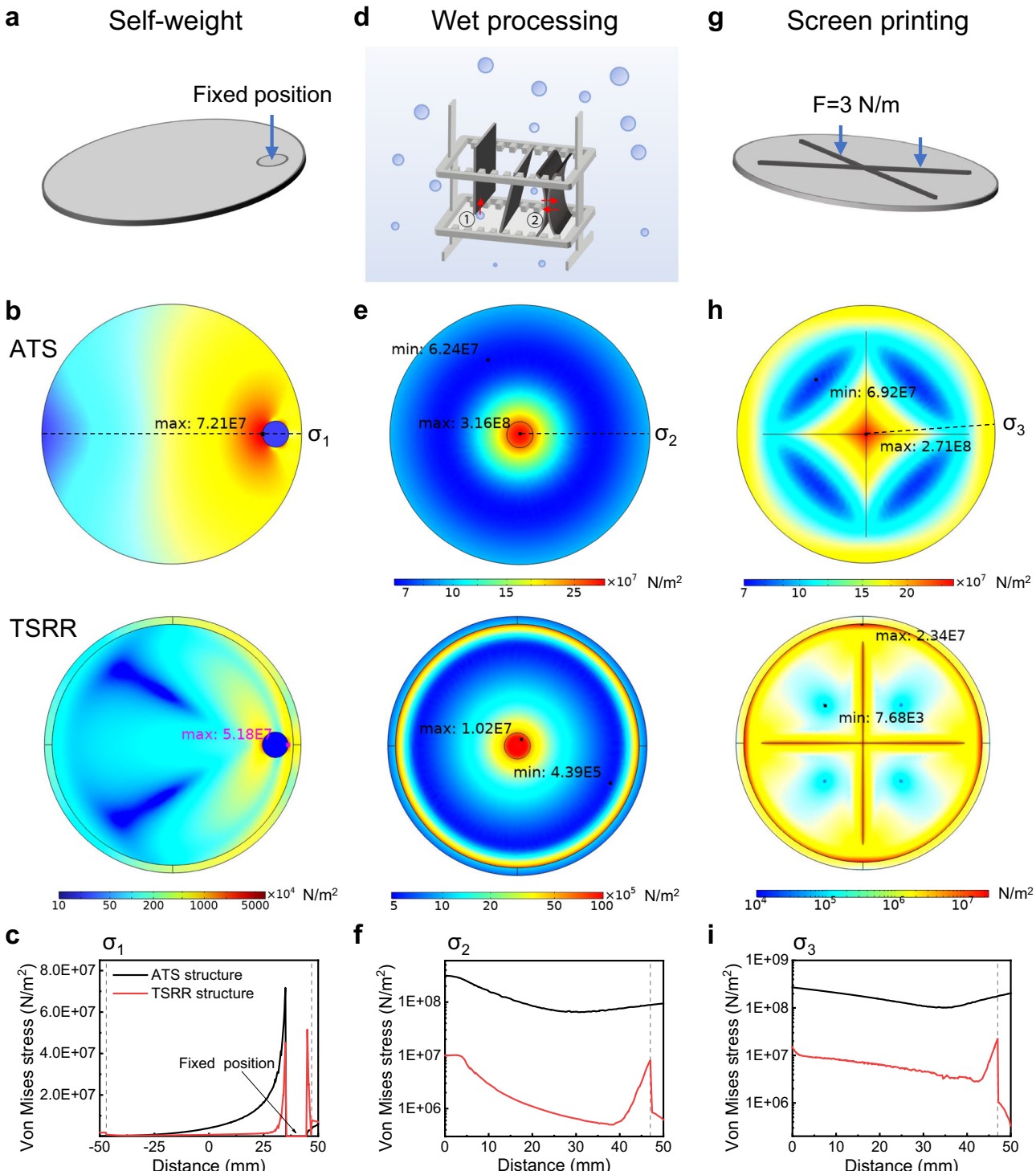

**Fig. 2 | Stress analysis for thin silicon wafers with ATS and TSRR structures in three cases during fabrication process in which the breakage rates are very high. a** Simplified schematic diagram of thin silicon wafer with a fixed position under the effect of gravity (corresponding to Fig. 1b, c), **b** Corresponding Von Mises stress profile of the ATS (top) and the TSRR (bottom) structures, and **c** the Von Mises stress distribution along the cut line $\sigma_1$ in (**b**). **d** Simplified schematic diagram of thin silicon wafers during wet processing, where, ①: Floatation due to buoyant force and bubbles. ②: Stiction due to surface tension. The corresponding **e** Von Mises stress profile of the ATS (top) and the TSRR (bottom) structures, and **f** Von Mises stress distribution along the cut line $\sigma_2$ in (**e**), where the central circular area with a radius of 5 mm is subjected to a total force of 0.2 N along the direction perpendicular to the surface of the wafer with the outermost edge of the wafers fixed. **g** Simplified schematic diagram of thin silicon wafer during screen printing, the corresponding **h** Von Mises stress profile of the ATS (top) and the TSRR (bottom) structures, and **i** Von Mises stress distribution along the cut line $\sigma_3$ in (**h**), where, crossed line loads of 3 N/m are applied with the outermost edge of the wafers fixed. The gray dashed lines in (**c**), (**f**) and (**i**) indicate the boundary between the reinforced ring and the central thin silicon region. Source data are provided as a Source Data file.

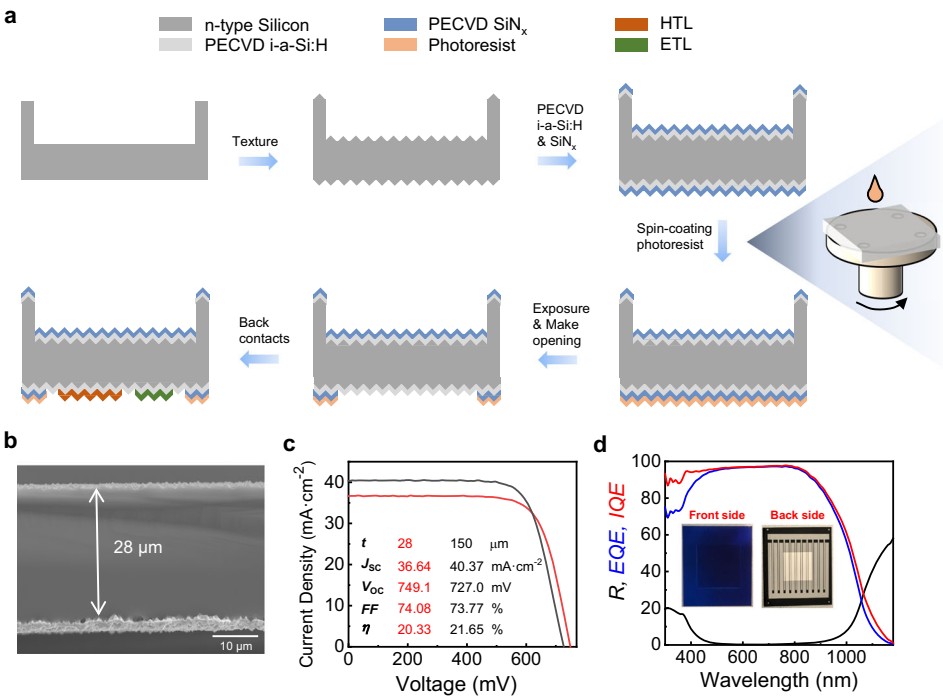

**Fig. 3 | Fabrication and performance of TSRR solar cells. a** Flow chart for free-standing thin silicon solar cells with all dopant-free and interdigitated back contacts using TSRR structure. **b** SEM image of the cross section of the champion thin silicon solar cell in our experiments. **c** Light *J-V* curves of the champion 28-μm thin cell and the 150-μm control sample. **d** The corresponding reflection (*R*), external quantum efficiency (*EQE*), and internal quantum efficiency (*IQE*) of the champion thin sample. The inset shows our real TSRR solar cell sample. Source data are provided as a Source Data file.

printing, which is probably the mechanical process with the highest probability of causing silicon breakage during the free-standing processing of thin silicon solar cells, since a large load from squeegee and adhesion force from metal paste will be applied to them[8]. Suppose that crossed line loads of 3 N/m are applied to the thin silicon wafer as shown in Fig. 2g, which is a simplified stress state of the thin silicon wafer under screen printing. The resulted Von Mises stress profile of the ATS structure (top) and the TSRR structure (bottom) are demonstrated in Fig. 2h (note the outermost edge of the wafers is fixed). The maximum and minimum stresses are 271.0 MPa and 69.2 MPa respectively for ATS structure and the maximum stress point is still in the center of the wafer. Notice that the stress is high near the outermost edge because it is fixed in the simulation. Concerning the TSRR structure, the maximum stress is an order of magnitude smaller than that of ATS structure, which is 23.4 MPa. And the minimum stress is only 0.00768 MPa. The maximum stress point located at the boundary between the reinforced ring and the central thin silicon region. A better comparison of the stress distribution along the cut line σ₃ in Fig. 2h is presented in Fig. 2i. Same to Fig. 2f, the tolerated stress of TSRR structure is always less than that of ATS structure and there is a steep increase in stress at the boundary of TSRR structure.

In addition, it is important to emphasize that, studies have shown that the cracking starts at the edge of the wafer and breakage occurs due to crack propagation[2,8,32], and Wieghold et al.[32], discovered that the critical force required to break a wafer decreases as thickness decreases based on their simulations of edge micro-crack propagation in wafers with different thicknesses, which implies the quality of the edges of the wafer is critical especially for thin wafer. Fortunately, our proposed thick reinforced ring of TSRR structure enables edges reinforcement of thin silicon wafer, which means that the reinforced ring not only shares the stress but also raise the critical force of breakage for thin silicon wafer.

## Fabrication of solar cells with TSRR structure

Using TSRR structure, we fabricated free-standing thin silicon solar cells with all dopant-free and interdigitated back contacts to confirm that this structure is suitable for solar cells. It is worth stating that the TSRR structure is applicable to any silicon technology such as passivated emitter and rear cell (PERC)[33], silicon heterojunction (SHJ)[34,35], tunnel oxide passivating contact (TOPCon)[36,37] as well as dopant-free passivating contact[38,39], and both front and back contacts (FBC) and interdigitated back contacts (IBC) structures[34,40]. While there are limitations to use those technologies on thin silicon wafers with substrate. Since one side of the thin wafer needs to be in contact with the substrate for support, the substrate becomes a barrier when processing this side, leading to complex preparation processes, such as removing the substrate on this side and attaching another carrier on the opposite side[12,16]. The flow chart in our experiments is given in Fig. 3a, the process began with the free-standing samples with TSRR structure etched from 150-μm thick silicon, n-type, single-crystalline wafers by the method shown in Fig. 1a. The thin samples were then textured and passivated. All dopant-free and back contacts were evaporated by a thermal evaporator at ambient temperature. Details of the process can be found in Methods section.

Figure 3b gives a SEM image of the cross section of the champion thin silicon solar cell in our experiments, its thickness is 28 μm and the height of the random pyramids is ∼ 2 μm. The light current density-voltage (*J-V*) curves of the champion thin cell and the 150-μm control sample are presented in Fig. 3c. A *J*ₛ𝒸 of 36.64 mA · cm⁻², a *V*ₒ𝒸 of 749.1 mV, an *FF* of 74.08% and an efficiency of 20.33% was achieved on 28-μm silicon solar cell. It is the highest efficiency reported for thin silicon solar cells with a thickness of <35 μm according to Table 1. The efficiency of the 150-μm control sample was 21.65% with a *J*ₛ𝒸 of 40.37 mA · cm⁻², a *V*ₒ𝒸 of 727.0 mV and an *FF* of 73.77%. The active areas of both of them are 1.007 cm² due to device limitations in our lab. Despite being more than four times thinner, the champion thin cell

**Table 2 | Numbers of sample breakage of ATS and TSRR structures in different processes**

| Structures | Thickness (µm) | Thinning | Texturing | Vacuuming[a] | Taping[b] | Others[c] | Total |
|---|---|---|---|---|---|---|---|
| ATS | 21 ~ 29 | 7(20) | 9(12) | 1(3) | 1(1) | 2 | 20 |
| | 51 ~ 57 | 1(10) | 1(8) | 0(6) | 0(6) | 2 | 4 |
| TSRR | 19 ~ 23 | 0(10) | 0(10) | 0(10) | 0(10) | 0 | 0 |
| | 42 ~ 50 | 0(10) | 0(10) | 0(10) | 0(10) | 0 | 0 |

[a]It is the vacuuming process during spin coating shown on the right-hand side in Fig. 3a.
[b]It is the process that taping the silicon samples to masks using tapes and then peel tapes off.
[c]Others including handling and transferring of samples.

achieves 93.9% of the efficiency of the original 150-µm thick silicon control sample. The main gain is a 22.1 mV boost in $V_{OC}$, which is attributed to a reduction in bulk recombination[41]. The 28-µm thin device achieved a certified efficiency of 20.05% at a credible third-party photovoltaic laboratory (Supplementary Fig. 6). The corresponding reflection ($R$), external quantum efficiency ($EQE$), and internal quantum efficiency ($IQE$) obtained for the champion thin sample are shown in Fig. 3d, and the inset shows the photo of our real solar cell with TSRR structure. The integrated current density $J_{EQE}$ extracted from the $EQE$ is 36.63 mA·cm$^{-2}$ and the total reflection is 6.38%. More details and improvement of this type of dopant-free IBC solar cell can be found in our previous works[39,42].

Meanwhile, numbers of sample breakage of the two structures during thinning, texturing, vacuuming, taping, handling and transferring processes were tracked, as shown in Table 2. As for the ATS structure, there were 20 samples in total for 21 ~ 29- µm group and 10 samples in total for 51 ~ 57-µm group. The area for all the samples was 2.4 × 2.8 cm$^2$. With respect to TSRR structure, for both 19 ~ 23-µm and 42 ~ 50-µm groups, the number of samples was 10. The whole area of these samples was also 2.4 × 2.8 cm$^2$ and the width of reinforced ring was 3 ~ 5.5 mm. The original thickness of the 50 samples mentioned above was 250 µm. Parentheses in the table represent the number of samples left at the beginning of this stage.

We start by looking at the first group of ATS structure, during the thinning process, i.e., thinning silicon from 250 µm to 21 ~ 29 µm with alkaline solution, 7 samples broke. Followed by 9 samples broke during texturing and 1 sample was fragmented during handling and transferring between the two processes. Surprisingly, the breakage rate was as high as 85% after just these two wet process treatments. We can conclude that the wet processing step is crucial for ultra-thin (<30 µm) silicon wafers, and we need to be careful in order to minimize the breakage rate. In some solar cell preparation processes, vacuuming and taping are required. The numbers of breakage for both processes were 1, but note that at the beginning of taping, there was only one sample left, as there was 1 sample breakage in handling and transferring between them. Therefore, it can be concluded that the breakage rate is >85% for fabricating 21 ~ 29 µm solar cells with an area of 2.4 × 2.8 cm$^2$ using ATS structure. Breakage rate can even increase to 100% if vacuuming and taping are required in the fabrication. As the thickness increases to 51 ~ 57 µm, the breakage rate decreases to 40% for ATS structure. In sharp contrast, the breakage rates of both 19 ~ 23-µm and 42 ~ 50-µm groups were 0% for TSRR structure. The aforementioned results prove that the reinforced ring of TSRR structure can greatly reduce the breakage rate during the preparation of thin silicon solar cells.

## Optoelectrical performance of solar cells with TSRR structure
To gain an in-depth understanding of the effect of TSRR structure on the optoelectrical performance of solar cells, we performed a TCAD numerical simulation[43]. And, in order to provide the readers with a better understanding, we simulated the FBC solar cells with TSRR structure instead of the IBC solar cells since the carrier transport mechanism of the FBC solar cells is simpler. We first investigated the direction of carrier transport of the solar cell with TSRR structure. We

call the width $W_1$ of the reinforced ring as a percentage of the width $W_2$ of the whole solar cell Ratio ($W_1/W_2$). For the purpose of providing a clearer picture of the direction of current density, we provide the simulated light current density map under AM1.5 G solar spectrum with Ratio = 10% in Fig. 4a, the inset shows the SHJ structure in the simulation (not scaled). The thicknesses of the central thin silicon region and the reinforced ring are 30 µm and 250 µm, respectively. Other parameters used in this simulation are given in Methods section. The direction of current transport represented by the gray arrows in the central thin silicon region is the same as that of conventional FBC solar cells. While the direction of current transport represented by the orange arrows is unique due to the presence of reinforced ring, since the photogenerated carriers generated at the reinforced ring region need to transport to the metal electrodes to be collected. This movement of carriers affects the $FF$ of solar cells.

The effect of the thickness of the central thin silicon region and Ratio on optoelectrical performance of solar cells are detailed in Fig. 4b–e. We all know that thicker silicon body absorbs more light, resulting in a larger $J_{SC}$. Therefore, as Ratio increases, i.e., the width of the 250-µm reinforced ring expands, more light can be absorbed and $J_{SC}$ grows, as depicted in Fig. 4b. Similarly, the bulk recombination is reduced as the thickness of silicon decreases, leading to a boost in $V_{OC}$. Consequently, $V_{OC}$ increases with decreasing the thickness of the central thin silicon region and Ratio as shown in Fig. 4c. As for $FF$, $FF$ increases as the thickness decreases since the longitudinal distance required for carrier transport to the electrodes decreases for conventional FBC solar cells. With regard to FBC solar cells with TSRR structure, as mentioned above, the photogenerated carriers in the reinforced ring region need to transport to the metal electrodes to be collected, and the number of these carriers is related to the width of the reinforced ring. As a result, when Ratio increases (0 <Ratio <25%), the number of these carriers that need to travel more distance before being collected increases, and thus the $FF$ decreases. But when Ratio is particularly large (Ratio > 30%), the resistance of longitudinal transport of these carriers decreases and $FF$ starts to increase, as exhibited in Fig. 4d. The trend of the resulted efficiency is presented in Fig. 4e. These results suggest to us that the value of Ratio should ideally be less than ~10% to maintain high efficiency. The above principles also apply to IBC solar cells with TSRR structure. Moreover, we provide comparison of the optoelectrical performance of the ATS and TSRR solar cells with both FBC and IBC structures as demonstrated in Supplementary Fig. 7 and Fig. 8 to give the readers a clearer picture of the TSRR solar cells.

## Industrial compatibility of TSRR structure
To validate the industrial compatibility of TSRR structure, we further prepared textured TSRR wafers and performed some key manufacturing processes for mass production of silicon solar cells based on 182 × 182 mm$^2$ pseudo-square wafers with an original thickness of 150 µm which are generally used in industry.

First, we prepared textured TSRR wafers starting from 182×182 mm$^2$ pseudo-square wafers with a thickness of 150 µm. For all TSRR wafers below, the thickness and width of the reinforced ring are 150 µm and 15 mm, respectively, i.e., the Ratio is 8.2%. According to our

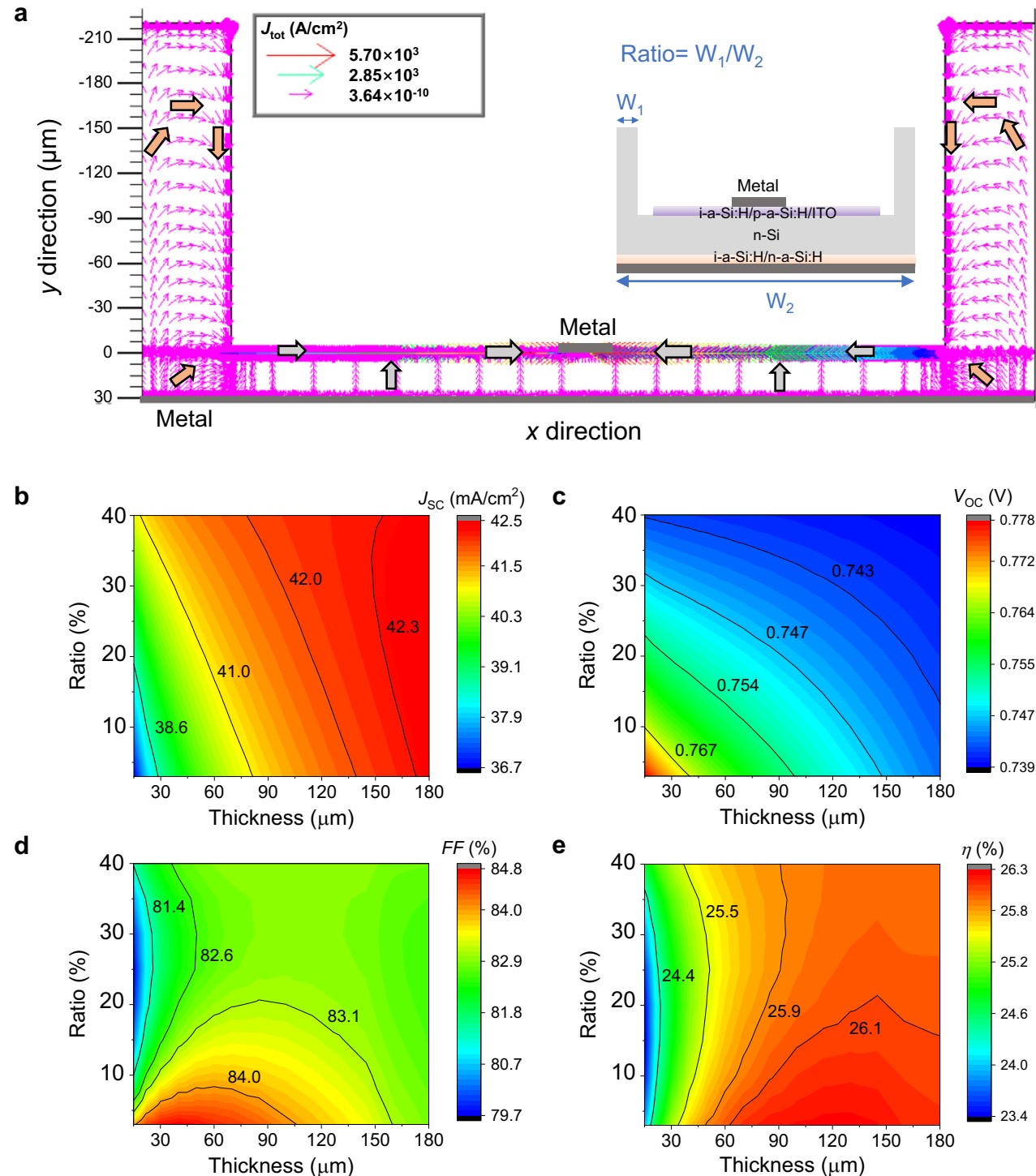

**Fig. 4 | Simulated optoelectrical performance of TSRR solar cells. a** Simulated light current density map under AM1.5 G solar spectrum when applied voltage is 0 with Ratio = 10%. Ratio is the width of the reinforced ring ($W_1$) as a percentage of the width of the whole solar cell ($W_2$). The inset shows the SHJ structure in the simulation (not scaled). The direction of current transport represented by the gray arrows in the central thin silicon region is the same as that of conventional FBC solar cells. While the direction of current transport represented by the orange arrows is unique due to the presence of reinforced ring. Calculated **b** $J_{SC}$, **c** $V_{OC}$, **d** $FF$ and **e** $\eta$ with varying the thickness of the central thin silicon region and Ratio. Source data are provided as a Source Data file.

experiments, the total breakage rate is 100% (20/20) for thinning the wafers with ATS structure from 150 μm to 40 μm and then cleaning by deionized water. In contrast, the total breakage rate is 0% (0/10) for preparing 40-μm TSRR wafers and then texturing. Figure 5a displays a 20-μm textured TSRR wafer and the thinnest textured silicon wafer we have successfully fabricated was as low as 14 μm.

Then we successfully performed screen printing at 170 °C using low-temperature silver paste on 60-μm textured TSRR wafers as shown in Fig. 5b. And its flexibility performance is demonstrated in Fig. 5c. The breakage rate during this process is 0% (0/5). We should admit that there are finger interruptions at the boundary of the reinforced ring and the central thin silicon region because of the steep slopes there. This may require further optimization of screen printing or

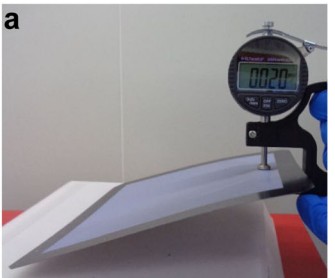
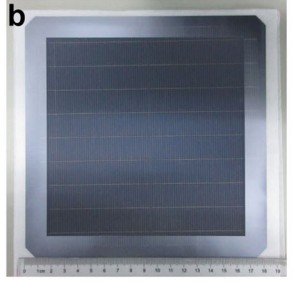
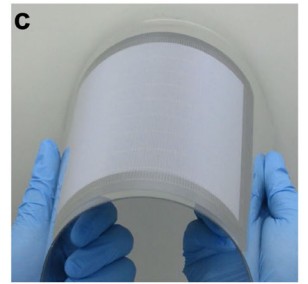
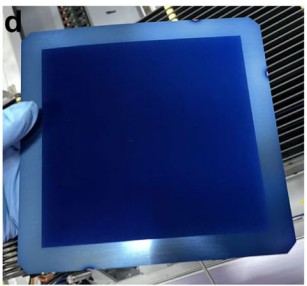

**Fig. 5 | Confirmation for industrial compatibility of TSRR structure. a** 20-μm textured wafer with TSRR structure. **b** The front side of a 60-μm textured TSRR wafer after screen printing and **c** its flexibility performance. **d** The 60-μm textured TSRR wafer after multiple high-temperature and wet manufacturing processes. Note, all used wafers are based on 182 × 182 mm$^2$ pseudo-square wafers with an original thickness of 150 μm.

**Table 3 | Numbers of sample breakage of 182 × 182 mm$^2$ pseudo-square wafers with 50 ~ 60-μm TSRR structure during multiple manufacturing processes**

| Steps | (1) | (2) | (3) | (4) | (5) | (6) | (7) | (8) | (9) | (10) | (11) |
|---|---|---|---|---|---|---|---|---|---|---|---|
| Breakages | 6 (57) | 0 (50) | 0 (50) | 2 (50) | 10 (48) | 0 (38) | 1 (25) | 9 (24) | 0 (15) | 0 (15) | 0 (15) |

Parentheses in the second line represent the number of samples left at the beginning of this process.

development of new metallization methods such as metal plating[44]. We think this is acceptable since this is the world's first attempt at a manufacturing process for this kind of thin structure, and there are still some processing details to be worked out together. What's more, if we prepare IBC solar cells whose back side is flat as shown in Fig. 3a, or if we decide to use this thin silicon structure with the reinforced ring cut off when it comes to applying it in some scenarios, the finger interruptions will be no longer an issue.

Last, using a total number of 57 pseudo-square wafers with textured TSRR structure with a thickness of 50 ~ 60 μm, we performed some high-temperature and wet manufacturing processes which are essential for mass production of silicon solar cells. Some pictures during these processes were recorded as shown in Supplementary Fig. 9. We also tracked the numbers of sample breakage in main process steps as shown in Table 3, and these steps include (1) previous RCA cleaning, (2) front-side boron diffusion, (3) back-side thermal SiO$_2$ at 1050 °C, (4) single-side SiO$_2$ removal, (5) alkali polishing, (6) p$^+$ doped poly-Si deposition at 430 °C and annealing at 900 °C, (7) front-side phosphosilicate glass (PSG) removal, (8) RCA cleaning, (9) Al$_2$O$_3$ deposition by atomic layer deposition (ALD) for surface passivation, (10) front-side and (11) back-side PECVD SiNx at 520 °C as ARC. The sample after going through all the above steps is shown in Fig. 5d. Note, ① after step (1), one sample broke when transferring; ② after step (6), 13 samples broke by accident, which should not be included in the breakage count. We can find that the breakage rate is 0% in these high-temperature processes, and most of the breakages occur in the wet processes, which is consistent with the results in Table 2. Breakage rate can be reduced by adjusting some operational details in these steps to make them more suitable for processing thin silicon.

According to above results, we can conclude that our TSRR method is industrial compatible.

## Discussion
In summary, we present a TSRR structure, which requires only 3 more steps with common devices in photovoltaic factories for mass production, and free-standing 4-inch 4.7-μm crystalline silicon wafer (Ratio ≈ 3%) was successfully prepared by this method. This is the largest area of free-standing monocrystalline silicon with a thickness of <5 μm reported so far based on our knowledge. Then with the help of COMSOL Multiphysics, we investigated the mechanical properties of TSRR structure and ATS structure under three cases, and the simulation results revealed that the reinforced ring of TSRR structure can

share a large stress when subjected to external forces, thus making the central thin silicon region of TSRR structure bear a smaller force compared to ATS structure. We further prepared solar cells with TSRR structure and obtained an efficiency of 20.33% (certified 20.05%) on 28-μm silicon solar cell with all dopant-free and interdigitated back contacts, which is the highest efficiency reported for thin silicon solar cells with a thickness of <35 μm. Meanwhile, the breakage rate of each process of solar cell fabrication with both structures were tracked. The results demonstrated that the breakage rate of 21 ~ 29-μm group with an area of 2.4 × 2.8 cm$^2$ was 85% ~100% for ATS structure, while the breakage rate of 19 ~ 23-μm group with a whole area of 2.4 × 2.8 cm$^2$ was 0% for TSRR structure. The above simulations and experiments confirmed that the reinforced ring can provide support throughout the solar cell preparation process and thus greatly suppressing the breakage rate. Then, based on TCAD numerical simulations, we investigated the carrier transport mechanism of the solar cell with TSRR structure, and the impact of thickness of the central thin silicon region and the width of the reinforced ring on the solar cell performance, which suggested that the value of Ratio should ideally be less than ~10% to maintain high efficiency. Finally, we prepared 50 ~ 60-μm textured TSRR wafers (Ratio = 8.2%) based on 182 × 182 mm$^2$ pseudo-square wafers with an original thickness of 150 μm, and then performed screen printing, high-temperature and wet manufacturing processes, which confirms the industrial compatibility of TSRR structure. We believe that this TSRR method is a feasible solution for the mass production of thin silicon solar cells.

## Methods
### Fabricating thin dopant-free IBC solar cells
The process began with the free-standing samples with TSRR structure etched from 150-μm thick silicon, n-type, single-crystalline wafers by the method shown in Fig. 1a. Then both sides of the samples were textured and covered symmetrically with 6 nm intrinsic amorphous silicon (i-a-Si:H) as passivation layer and 85 nm SiN$_x$ as antireflection layer by PECVD. The photoresist was subsequently spin-coated on the backside of the samples. Note that the air holes for vacuuming are designed to be under the reinforced ring rather than the central thin silicon region to reduce the breakage rate, as shown in the illustration on the right in Fig. 3a. Next, MoO$_x$ (10 nm)/Ag (300 nm) and LiF$_x$ (1 nm)/Al (400 nm) films were deposited in the area exposed by ultraviolet (UV) light by thermal evaporation at ambient temperature with metal shadow masks to serve as hole-transport layer (HTL) and

**Table 4 | Parameters used for the simulation**

| Parameters | n-Si | i-a-Si:H | p-a-Si:H | n-a-Si:H |
|---|---|---|---|---|
| Electron affinity (eV) | 4.05 | 3.82 | 3.82 | 3.82 |
| Bandgap energy (eV) | 1.12 | 1.75 | 1.75 | 1.75 |
| Doping concentration (cm$^{-3}$) | $2 \times 10^{15}$ | $1 \times 10^{15}$ | $6 \times 10^{19}$ | $1 \times 10^{19}$ |
| Layer thickness | 15 ~ 250 µm | 6 nm | 10 nm | 10 nm |
| SRH lifetime | 6 ms | 10 µs | 10 µs | 10 µs |

electron-transport layer (ETL), respectively. The widths of the HTL and ETL were 1225 µm and 625 µm, respectively, and the space between them was 150 µm.

## TCAD simulation

The simulation parameters can be found in Table 4. Shockley-Read-Hall (SRH) recombination, Auger recombination, Fermi-Dirac carrier statistics and bandgap narrowing model are deployed into the simulation.

## Reporting summary

Further information on this research is available in the Nature Portfolio Reporting Summary linked to this article.

## Data availability

All data generated or analyzed during this study are included in the published article and its Supplementary Information and Source Data files. Source data are provided with this paper.

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

## Acknowledgements

This work was supported by the Major State Basic Research Development Program of China (Grant No. 2022YFB4200101, W. S.), the National Natural Science Foundation of China (Grant No. 11974242, 11834011, 62034009, 62104268, W. S., W. S., P. G. and H. L.), Inner Mongolia Science and Technology Project (Grant No. 2022JBGS0036, W. S.), Shenzhen Fundamental Research Program (Grant No. JCYJ20200109142425294, P. G.) and Guangdong Basic and Applied Basic Research Foundation (Grant No. 2019B151502053, P. G.).

## Author contributions

H. L., P. G. and W. S. supervised the work. H. L., T. W. and Z. L. conducted the idea and designed the experiments. T. W. conducted the COMSOL and TCAD simulations. T. W. and Z. L. conducted most of the experiments and prepared the manuscript. H. L. assisted with wafers preparation and devices fabrication. H. L., P. G. and W. S. contributed to the revision of the manuscript.

## Competing interests

The authors declare no competing interests.
