## [Peer Review File · Nature Communications]

Free-standing ultrathin silicon wafers and solar cells through edges reinforcementREVIEWER COMMENTS

Reviewer #1 (Remarks to the Author):

This paper provide a new approach of ultra-thin crystalline silicon solar cell, the proposal is clear and interesting, however, the process and results are not satisfied, some comments and suggestion are listed below

- 1.the authors describe that the approach is “saving of silicon raw material”, “low-cost”, but the process start from 225 um double-polished silicon wafer, and most of the material is etched and wasted, the thickness is almost double of conventional crystalline silicon solar cell, the authors should explain this issue and compare with other approaches, such as epitaxial growth, left-off of ion implantation, etc.
- 2.in the preparation of TSRR structure, do you measure the thickness uniformity? The thickness uniformity should be shown.
- 3.in the fabrication of solar cells with TSRR structure, the authors say that is applicable to any silicon technologies such as PERC, TOPCon, etc., but, in the paper, the authors only show a IBC process and the metal contact is formed with vacuum evaporation. Please show how to fabricate metal contacts on both sides of the wafer? Is it possible to apply screen printing on both sides? Otherwise, this technology is not Industrial compatibility.
- 4.in figure 3, the J-V curves are shown, the authors announced “the highest efficiency reported for thin silicon solar cells with a thickness of $< 34 \mu\text{m}$ ”, I suggest a third-party measurement should be done and the report should be uploaded.
- 5.the front and back photos of real solar cells with TSRR structure should be shown in figure 3 together with the cross-sectional SEM image, or supply the photos of cells in the supporting information.
- 6.in the “performance of solar cells with TSRR”, the authors analysis the electrical performances with TACD simulation method, however, the mode of cell structure is not clear and is very qualitative, the necessity here is not felt, the simulation should base on the results of previous sessions.

Reviewer #2 (Remarks to the Author):

In this manuscript, the reinforcement ring is proposed to enhance the mechanical properties of thin silicon wafers, making it accessible to greatly reduce the breakage rate during the preparation of thin silicon solar cells. It is impressive that the breakage rate can be reduced to 0% for free-standing edge-reinforced thin solar cells compared to $>85\%$ for normal thin solar cells and the reported efficiency of the 31- μm silicon solar cell has reached up to 20.4%. What’s more, the flexibility of the edge-reinforced thin solar cells is preserved and is in control. The manuscript is also well organized, from display of a new thin silicon structure with reinforcement ring and its mechanical properties analysis to preparation of thin

silicon solar cell with reinforcement ring and its photoelectric properties analysis, providing us a comprehensive understanding of edge-reinforced thin solar cells. But I have several concerns,

1. As shown in Fig 1a, why is SiNx in the backside left when thinning by alkaline solution. That is, why not make opening on both sides? It would be faster to perform a double-sided etching.
2. Why did you choose a force of 0.2 N in Fig 2d instead of other values, what is the basis for the selection of this value? Similarly, why did you set 3 N/cm in Fig 2g?
3. It is hard to understand that the maximum stress point is deviated from the center of the wafer for edge-reinforced structure in Fig 2e. Can you explain that?
4. Can you explain why the solar cell with a thickness of 225 μm in Figure 3c has such a low VOC?
5. In Fig 4b-e, only the effect of thickness and Ratio on optoelectrical performance of the edge-reinforced solar cells is illustrated, and it is better to provide a comparison of the optoelectrical performance of the edge-reinforced and normal solar cells with same parameters to give the readers a clearer picture of the edge-reinforced solar cells.
6. Round silicon wafer is used to illustrate the role of the reinforcement ring in this manuscript, is reinforcement ring suitable for square wafers used in silicon solar cell production lines?

Reviewer #3 (Remarks to the Author):

What are the noteworthy results?

The authors report on a method to fabricate thin silicon solar cells where the mechanical integrity of the sample is provided by a thick edge. This thick edge makes the cell compatible with the conventional solar cell manufacturing technologies.

Will the work be of significance to the field and related fields?

How does it compare to the established literature? If the work is not original, please provide relevant references.

This approach has been suggested and demonstrated previously for space applications as well as commercially proposed by the company Cubic PV (named 1366 at the time, <https://www.greentechmedia.com/articles/read/1366-technologies-unveils-new-wafer-technology>). This paper focusses on round wafers, while the whole industry uses (pseudo)-square wafers. The authors would have to demonstrate their method as well on square wafers for maximum impact. They should

also comment on the techno-economic aspects, i.e., the conventional gains in terms of the reduction in silicon costs are not there and what are the expected costs for the thinning process. Also it would be good if the authors could show some high-temperature processes for their samples as they are now commonly used for PERC and TOPCon solar cells.

Does the work support the conclusions and claims, or is additional evidence needed?

Yes, the work supports the conclusions and claims.

Are there any flaws in the data analysis, interpretation and conclusions? Do these prohibit publication or require revision?

No, the data analysis etc. is done very well and support their conclusions.

Is the methodology sound? Does the work meet the expected standards in your field?

Yes.

Is there enough detail provided in the methods for the work to be reproduced?

Yes.

Response to Reviewers' comments on NCOMMS-23-36383

Reviewer #1:

This paper provide a new approach of ultra-thin crystalline silicon solar cell, the proposal is clear and interesting.

Reply: We are happy to have this positive recommendation from the reviewer.

however, the process and results are not satisfied, some comments and suggestion are listed below

1. the authors describe that the approach is “saving of silicon raw material”, “low-cost”, but the process start from 225 um double-polished silicon wafer, and most of the material is etched and wasted, the thickness is almost double of conventional crystalline silicon solar cell, the authors should explain this issue and compare with other approaches, such as epitaxial growth, left-off of ion implantation, etc.

Reply: We thank the reviewer very much for his/her careful review of the manuscript. We apologize for any misunderstanding caused by our unclear descriptions. We have removed such descriptions. Instead, we use “industrially compatible” to describe our approach. We also demonstrate TSRR process starting from 150 μm pseudo-square wafers which are generally used in industry, as shown in Question 3 from Reviewer #1. As for cost issues, please refer to Question 3 from Reviewer #3.

2. in the preparation of TSRR structure, do you measure the thickness uniformity? The thickness uniformity should be shown.

Reply: Thanks for this question and suggestion. Actually, we can see its thickness uniformity from the color uniformity under white light illumination shown in Figure 1d. As for quantitative uniformity measurements, 11 points in a 4-inch wafer with TSRR structure were further measured as shown in the figure below. Their thicknesses are from 5.5 to 11.1 μm , i.e., the total thickness variation (TTV) is within 6 μm .

Figure S2. (a) The quantitative thickness uniformity measurement results in a 4-inch wafer with TSRR structure and (b) the corresponding SEM images (in order from thin to thick) of the cross section for each point in (a).

We have made some changes in the revised manuscript (page 7) to provide the quantitative thickness uniformity measurements:

“At the same time, based on the color uniformity shown here, we can also see that the thickness of the wafer is fairly uniform, and according to our further quantitative measurements on the thickness uniformity, as demonstrated in Figure S2, the total thickness variation (TTV) for this thin silicon preparation method is within 6 μm.”

3. in the fabrication of solar cells with TSRR structure, the authors say that is applicable to any silicon technologies such as PERC, TOPCon, etc., but, in the paper, the authors only show a IBC process and the metal contact is formed with vacuum evaporation. Please show how to fabricate metal contacts on both sides of the wafer? Is it possible to apply screen printing on both sides? Otherwise, this technology is not Industrial

compatibility.

Reply: Thanks for this constructive suggestion. As shown below, we prepared textured TSRR wafers and performed screen printing based on $182 \times 182 \text{ mm}^2$ pseudo-square wafers with an original thickness of $150 \text{ }\mu\text{m}$ which are generally used in industry. In addition, we further performed some high-temperature and wet manufacturing processes as illustrated in Question 4 from Reviewer #3.

First, we prepared textured TSRR wafers starting from $182 \times 182 \text{ mm}^2$ pseudo-square wafers with a thickness of $150 \text{ }\mu\text{m}$. For all TSRR wafers below, the thickness and width of the reinforced ring are $150 \text{ }\mu\text{m}$ and 15 mm , respectively, i.e., the Ratio is 8.2%. According to our experiments, the total breakage rate is 100% (20/20) for thinning the wafers with ATS structure from $150 \text{ }\mu\text{m}$ to $40 \text{ }\mu\text{m}$ and then cleaning by deionized water. In contrast, the total breakage rate is 0% (0/10) for preparing $40\text{-}\mu\text{m}$ TSRR wafers and then texturing. Figure (a) displays a $20\text{-}\mu\text{m}$ textured TSRR wafer and the thinnest textured silicon wafer we have successfully fabricated was as low as $14 \text{ }\mu\text{m}$.

Then we successfully performed screen printing at $170 \text{ }^\circ\text{C}$ using low-temperature silver paste on both front side (left) and back side (right) of a $60\text{-}\mu\text{m}$ textured TSRR wafer as shown in Figure (b). And its flexibility performance is demonstrated in Figure (c). The breakage rate during this process is 0% (0/5). We should admit that there are finger interruptions at the boundary of the reinforced ring and the central thin silicon region because of the steep slopes there. This may require further optimization of screen printing or development of new metallization methods such as metal plating¹. We think this is acceptable since this is the world's first attempt at a manufacturing process for this kind of thin structure, and there are still some processing details to be worked out together. What's more, if we prepare IBC solar cells whose back side is flat as shown in Figure 3a, or if we decide to use this thin silicon structure with the reinforced ring cut off when it comes to applying it in some scenarios, the finger interruptions will be no longer an issue.

(a) $20\text{-}\mu\text{m}$ textured wafer with TSRR structure. (b) The front side (left) and back side (right) of a $60\text{-}\mu\text{m}$ textured TSRR wafer after screen printing, and (c) its flexibility

performance.

we have added the above content to the last subsection “**Industrial compatibility of TSRR structure**” in the manuscript to validate the industrial compatibility of TSRR structure. Please see changes in the manuscript (page 16).

4. in figure 3, the J-V curves are shown, the authors announced “the highest efficiency reported for thin silicon solar cells with a thickness of < 34 μm”, I suggest a third-party measurement should be done and the report should be uploaded.

Reply: Thanks for careful review of the manuscript. We previously consulted with a third-party measurement center in China, but it was difficult to get our samples certified because there is no suitable holder for our small and ultrathin samples. And we have removed “it is the highest efficiency reported for thin silicon solar cells with a thickness of < 34 μm” such description since we haven’t certified it.

5.the front and back photos of real solar cells with TSRR structure should be shown in figure 3 together with the cross-sectional SEM image, or supply the photos of cells in the supporting information.

Reply: Thanks for this suggestion. We have provided the photo of our thin solar cell with TSRR structure sample in Figure 3d, please see changes below and in the manuscript.

6.in the “performance of solar cells with TSRR”, the authors analysis the electrical performances with TACD simulation method, however, the mode of cell structure is not clear and is very qualitative, the necessity here is not felt, the simulation should base on the results of previous sessions.

Reply: We thank the reviewer very much for his/her careful review of the manuscript. Since the solar cell with TSRR structure is new to all of us, here we provide its carrier transport mechanism, and the effect of thickness and Ratio (the width of the reinforced ring as a percentage of the width of the whole solar cell) on the cell performances to give the readers a clear picture of the TSRR solar cells. Only when we understand how it works can we design the structure to be more efficient.

The mode of structure is based on a conventional SHJ structure but with reinforced ring shown in the inset in Figure 4a, and the Ratio is 10%, and the other parameters in simulation are described in the “Performance of solar cells with TSRR” and “Methods” subsections. The efficiency (20.4%) described in the previous subsection (“Fabrication of solar cells with TSRR structure”) do not take into account the reinforced ring (only the central thin silicon region of TSRR structure is illuminated), thus this efficiency is based on conventional IBC solar cells. But when it comes to future applications, we should decide whether to cut the reinforced ring based on specific application situations. Therefore, we believe that it is necessary to introduce the basic optoelectrical mechanism of the TSRR solar cell.

Reviewer #2:

In this manuscript, the reinforcement ring is proposed to enhance the mechanical properties of thin silicon wafers, making it accessible to greatly reduce the breakage rate during the preparation of thin silicon solar cells. It is impressive that the breakage rate can be reduced to 0% for free-standing edge-reinforced thin solar cells compared to >85% for normal thin solar cells and the reported efficiency of the 31- μ m silicon solar cell has reached up to 20.4%. What’s more, the flexibility of the edge-reinforced thin solar cells is preserved and is in control. The manuscript is also well organized, from display of a new thin silicon structure with reinforcement ring and its mechanical properties analysis to preparation of thin silicon solar cell with reinforcement ring and its photoelectric properties analysis, providing us a comprehensive understanding of edge-reinforced thin solar cells.

Reply: We thank the reviewer very much for his/her careful review of the manuscript, and we are very happy to have this positive recommendation from the reviewer.

But I have several concerns,

1. As shown in Fig 1a, why is SiNx in the backside left when thinning by alkaline solution. That is, why not make opening on both sides? It would be faster to perform a double-sided etching.

Reply: Thanks for this question. The choice of one-sided or double-sided etching depends on the subsequent fabricating process for thin silicon solar cells. As shown in Figure 3(a), the thin samples were only front-sided textured after the thinning process due to protection from LPCVD SiNx on the backside. Since it is better to make interdigitated back contacts (IBC) on flat surfaces than on textured surfaces, we opted for only one-sided etching. If we are going to prepare bifacial thin silicon solar cells,

we no longer need LPCVD SiNx to protect our backside from being textured, so we can choose double-sided etching which is faster than one-sided etching.

2. Why did you choose a force of 0.2 N in Fig 2d instead of other values, what is the basis for the selection of this value? Similarly, why did you set 3 N/cm in Fig 2g?

Reply: Thanks for careful review of the manuscript. These values are set so that the resulting maximum value of Von Mises stress of the ATS structure is around the fracture strength of silicon (80~520 MPa). It is meaningful to explore the stress distribution close to the critical state of breakage, because it can help us better understand the reasons why TSRR structure can reduce breakage rate.

3. It is hard to understand that the maximum stress point is deviated from the center of the wafer for edge-reinforced structure in Fig 2e. Can you explain that?

Reply: Thanks for careful review of the manuscript. In some cases, due to the stress-sharing effect of the reinforced ring, the maximum stress points are slightly deviated from the center of the wafer (such as the case shown in Fig 2e), or even deviated severely and are at the boundary between the reinforced ring and the central thin silicon region (such as the case shown in Fig 2h), which was confirmed experimentally. As shown in the screenshot below, which comes from the Supporting video. The TSRR sample breaks initiate at the boundary since the maximum stress point locates at the boundary.

4. Can you explain why the solar cell with a thickness of 225 μm in Figure 3c has such a low V_{oc} ?

Reply: Thanks for this question. There are 2 main reasons for this:

(1) The implied V_{oc} (iV_{oc}) of the 225- μm sample is only $\sim 723\text{mV}$ due to the lack of optimization for passivation process (PECVD i-a-Si:H).

(2) Edge recombination caused by inevitable overlap between the hole- and electron-

transport layers deposited by thermally evaporation with metal masks^{2,3}, which affect V_{oc} and FF of silicon solar cells.

5. In Fig 4b-e, only the effect of thickness and Ratio on optoelectrical performance of the edge-reinforced solar cells is illustrated, and it is better to provide a comparison of the optoelectrical performance of the edge-reinforced and normal solar cells with same parameters to give the readers a clearer picture of the edge-reinforced solar cells.

Reply: Thanks for this suggestion, it is really helpful. We have added comparison of the optoelectrical performance of the ATS and TSRR solar cells with both FBC and IBC structures as shown below (we present them in Figure S6 and S7).

Figure S6. (a) The simulated ATS and TSRR structures with front and back contacts (FBC). (b) J_{sc} , (c) V_{oc} , (d) FF , and (e) η as a function of thickness for ATS and TSRR structures.

As shown in Figure S6a, it is the simulated optoelectrical performance of the ATS and TSRR solar cells with FBC structure. The simulation parameters can be found in Table 4 (manuscript) and the thickness of the reinforced ring and the Ratio for TSRR structure are 250 μm and 5%, respectively. Let's look into Figure S6b, when the thickness is 250 μm, the J_{sc} of ATS and TSRR structures are the same since the reinforced ring of TSRR structure is also 250 μm. J_{sc} decreases as the thickness decreases because thinner silicon body absorbs less light. Besides, the J_{sc} of TSRR structure is always larger than that of ATS structure due to the presence of thicker reinforced ring in TSRR structure. Next, let's move on to Figure S6c. V_{oc} increases as the thickness decreases due to reduced bulk recombination. Similarly to the J_{sc} , the V_{oc} of TSRR structure is always smaller than that of ATS structure due to more bulk recombination in the thick reinforced ring of TSRR structure. As for FF , when the thickness is 250 μm, there is a

0.2% difference between the two structures attributed to the difference in mesh settings in the simulation (the results shown in Figure S7 is more accurate because that simulation is based on the contacts being all on the backside), as indicated in Figure S6d. As thickness decreases, FF increases for ATS structure since the longitudinal distance required for carrier transport to the electrodes decreases. (Note that if the thickness is particularly low ($<10\ \mu\text{m}$), it will cause the FF to decrease, because the resistance of lateral transport of carriers to the front electrodes increases at that point). While for TSRR structure, (as mentioned in the manuscript) the photogenerated carriers in the reinforced ring region need to transport to the metal electrodes to be collected, which affect the FF , leading to a lower FF for the TSRR structure than the ATS structure, and FF begins to decrease at around $50\ \mu\text{m}$. The final η for both structures are demonstrated in Figure S6e.

Figure S7. (a) The simulated ATS and TSRR structures with interdigitated back contacts (IBC). (b) J_{sc} , (c) V_{oc} , (d) FF , and (e) η as a function of thickness for ATS and TSRR structures.

6. Round silicon wafer is used to illustrate the role of the reinforcement ring in this manuscript, is reinforcement ring suitable for square wafers used in silicon solar cell production lines?

Reply: Please go back to Question 3 from Reviewer #1. We have illustrated square wafers with TSRR structure.

Reviewer #3:

What are the noteworthy results?

The authors report on a method to fabricate thin silicon solar cells where the mechanical integrity of the sample is provided by a thick edge. This thick edge makes the cell compatible with the conventional solar cell manufacturing technologies.

Reply: We thank the reviewer very much for his/her careful review of the manuscript, and we are happy to have this positive recommendation from the reviewer.

Will the work be of significance to the field and related fields?

How does it compare to the established literature? If the work is not original, please provide relevant references.

1. This approach has been suggested and demonstrated previously for space applications as well as commercially proposed by the company Cubic PV (named 1366 at the time, <https://www.greentechmedia.com/articles/read/1366-technologies-unveils-new-wafer-technology>).

Reply: We thank the reviewer very much for his/her careful review of the manuscript and admire for his/her vast knowledge.

1366 is a multi-crystalline silicon wafer growing technology which forms a wafer directly from molten silicon in a bath-like furnace, with the ability to locally control wafer thickness. Thus, it can produce thin wafer with thick edge.

This idea and technology are innovative, but according to a report⁴ published in December 2016 from 1366 Technologies, Inc. (very little about this 1366 technology has been reported after 2016), it suffered from the following 4 serious problems:

1. Low bulk lifetime. Their latest generations of 1366 Direct Wafer usually show bulk lifetimes of 40 to 80 μ s measured at 1×10^{14} cm^{-3} excess carrier density. This is much smaller than the bulk lifetime of >1 ms of silicon produced by Czochralski Method, which is the most widely used and cost-effective method today for growing monocrystalline silicon. Low bulk lifetime can lead to a decrease in the efficiency of the solar cell.

2. High ratio of the thickness in the central thin region to the thickness in the edge region. They achieved a ratio value of 67% in 2016, i.e., the average thickness in the central thin region is 160 μ m and edge thickness is 240 μ m. In contrast, we have achieved a ratio of $< 3\%$, i.e., the thickness in the central thin region is 4.7 μ m shown in Figure 1b and the edge thickness is 192 μ m shown in Figure S1.

3. High total thickness variation (TTV) in the central thin region. In 2016, they

improved the process but only achieved a TTV value of 42% compared to the average thickness in the central thin region, i.e., the TTV is high to 65 μm when the average thickness in the central thin region is 155 μm . In contrast, for our samples, the TTV is within 6 μm illustrated in Question 2 from Reviewer #1.

4. Difficulty in growing very thin framed wafers. In their report, the central thin regions of the reported samples are all $\geq 85 \mu\text{m}$. As for the technique we report in this manuscript, it was successfully used to prepare 4.7- μm silicon wafer with 192- μm thick edge.

2. This paper focusses on round wafers, while the whole industry uses (pseudo)-square wafers. The authors would have to demonstrate their method as well on square wafers for maximum impact.

Reply: Please go back to Question 3 from Reviewer #1. We have illustrated square wafers with TSRR structure.

3. They should also comment on the techno-economic aspects, i.e., the conventional gains in terms of the reduction in silicon costs are not there and what are the expected costs for the thinning process cost.

Reply: Thanks for this suggestion. Same to Question 1 from Reviewer #1, we should admit that our method can not reduce cost but it is industrially compatible. Mass production for TSRR structure requires only 3 more steps:

1. Depositing SiNx on both sides of normal silicon wafer by PECVD/LPCVD.
2. Removing SiNx from the central region to make opening by laser.
3. Etching silicon to desired thickness using KOH solution.

Since PECVD/LPCVD, laser and KOH solution are routine in photovoltaic factories, the cost of these 3 steps is manageable, and the total cost of such edge-reinforced thin silicon solar cells is close to that of normal thickness silicon solar cells.

We believe that this technique proposed in this manuscript is a feasible solution for the mass production of thin silicon solar cells, which can solve the current problem of very high breakage rate. More importantly, lightweight and flexible thin silicon solar cell has a much wider range of application scenarios than conventional thickness silicon solar cell, so we can accept a wider range of price for thin solar cells.

4. Also it would be good if the authors could show some high-temperature processes for their samples as they are now commonly used for PERC and TOPCon solar cells.

Reply: Thanks for this constructive suggestion. In addition to screen printing process as shown in Question 3 from Reviewer #1, we have also performed some high-

temperature and wet manufacturing processes. We also added the following content to the last subsection “**Industrial compatibility of TSRR structure**” in the manuscript. Please see changes in the manuscript (page 16).

Last, using a total number of 57 pseudo-square wafers with textured TSRR structure with a thickness of 50~60 μm , we performed some high-temperature and wet manufacturing processes which are essential for mass production of silicon solar cells. Some pictures during these processes were recorded as shown in Figure S8. We also tracked the numbers of sample breakage in main process steps as shown in Table 3, and these steps include (1) previous RCA cleaning, (2) front-side boron diffusion, (3) back-side thermal SiO_2 at 1050 $^\circ\text{C}$, (4) single-side SiO_2 removal, (5) alkali polishing, (6) p^+ doped poly-Si deposition at 430 $^\circ\text{C}$ and annealing at 900 $^\circ\text{C}$, (7) front-side phosphosilicate glass (PSG) removal, (8) RCA cleaning, (9) Al_2O_3 deposition by atomic layer deposition (ALD) for surface passivation, (10) front-side and (11) back-side PECVD SiN_x at 520 $^\circ\text{C}$ as ARC. The sample after going through all the above steps is shown in Figure S8f. Note, ① after step (1), one sample broke when transferring; ② after step (6), 13 samples broke by accident, which should not be included in the breakage count. We can find that the breakage rate is 0% in these high-temperature processes, and most of the breakages occur in the wet processes, which is consistent with the results in Table 2. Breakage rate can be reduced by adjusting some operational details in these steps to make them more suitable for processing thin silicon.

According to above results, we can conclude that our TSRR method is industrial compatible.

Figure S8. Pictures of samples after step (a) 1, (b) 2, (c) 3, (d) 6, (e) 9 and (f) 11.

Table 3. Numbers of sample breakage of $182 \times 182 \text{ mm}^2$ pseudo-square wafers with 50~60- μm TSRR structure during multiple manufacturing processes.

Steps	(1)	(2)	(3)	(4)	(5)	(6)	(7)	(8)	(9)	(10)	(11)
Breakages	6 (57)	0 (50)	0 (50)	2 (50)	10 (48)	0 (38)	1 (25)	9 (24)	0 (15)	0 (15)	0 (15)

Note: Parentheses in the second line represent the number of samples left at the beginning of this process.

5. Does the work support the conclusions and claims, or is additional evidence needed?

Yes, the work supports the conclusions and claims.

Are there any flaws in the data analysis, interpretation and conclusions? Do these prohibit publication or require revision?

No, the data analysis etc. is done very well and support their conclusions.

Is the methodology sound? Does the work meet the expected standards in your field?

Yes.

Is there enough detail provided in the methods for the work to be reproduced?

Yes.

Reply: Thanks for careful review of the manuscript and we are very happy to have this positive recommendation from the reviewer.

The revisions have been tracked in the revised manuscript. We hope that the revised manuscript can be up to your requirements. Thank you very much for your reconsiderations.

References:

1. Lennon, A., Yao, Y., Wenham, S. Evolution of metal plating for silicon solar cell metallisation. *Prog. Photovolt. Res. Appl.* **21**, 1454-1468 (2012).
2. Lin, H., Wang, J., Wang, Z., Xu, Z., Gao, P., Shen, W. Edge effect in silicon solar cells with dopant-free interdigitated back-contacts. *Nano Energy* **74**, 104893 (2020).
3. Liu, Z. et al. Dual Functional Dopant-Free Contacts with Titanium Protecting Layer: Boosting Stability while Balancing Electron Transport and Recombination Losses. *Adv. Sci.* **9**, 2202240 (2022).
4. Lorenz, A. 1366 Project Automate: Enabling Automation for <\$0.10/W High-Efficiency Kerfless Wafers Manufactured in the US. 2017. <https://doi.org/10.2172/1356280>

REVIEWERS' COMMENTS

Reviewer #1 (Remarks to the Author):

The authors have replied all the comments and revised the manuscript, most of these responses and revisions are satisfied. But, I have further comments,

1. In the report Reviewer #1, comment No. 4, the authors replied, "We previously consulted with a third-party measurement center in China, but it was difficult to get our samples certified because there is no suitable holder for our small and ultrathin samples. And we have removed "it is the highest efficiency reported for thin silicon solar cells with a thickness of $< 34 \mu\text{m}$ " such description since we haven't certified it".

I strongly suggest the authors give the certificated report for the reasons: (1) the cells will be applied without the frame, it should be measured at the final application form without frame; (2) the values measured with frame are not correct because of the contribution and influences from the surrounding; (3) the cells can be easily measured at a third-party if they are cut off from the frame and the holder is supplied together with the cells.

2. I suggest the authors remove "high-success' in the title, because this technology has not been tested by the industry.

Reviewer #2 (Remarks to the Author):

See Attachment

After the carefully revision, I recommend this manuscript to the Nature Communications.

Reviewer #3 (Remarks to the Author):

Overall, I am happy with the answers provided by the authors on the issues raised in the first review, particularly related to the additional experimental work performed on square wafers and industry-relevant processes.

However, I am not yet happy with how prior art is addressed. The work from 1366 should also be cited in the manuscript instead of only in the rebuttal letter. In addition, this paper, Thin HI-ETA/sup (R)/ space silicon solar cells with improved end-of-life performance | IEEE Conference Publication | IEEE Xplore, should be discussed. I could not find the reference at the time of the first review, but I referred to it as work in the space silicon area. I will leave it up to the editor to assess how this affects the novelty of this work.

In addition, there are several typos/grammar mistakes in the revised version of the manuscript (e.g., rete instead of rate) that should be addressed.

Response to Reviewers' comments on NCOMMS-23-36383A

Reviewer #1:

The authors have replied all the comments and revised the manuscript, most of these responses and revisions are satisfied.

Reply: We thank the reviewer again for his/her careful review and constructive suggestions for our manuscript, and we are happy to have this positive recommendation.

But, I have further comments.

1. In the report Reviewer #1, comment No. 4, the authors replied, "We previously consulted with a third-party measurement center in China, but it was difficult to get our samples certified because there is no suitable holder for our small and ultrathin samples. And we have removed "it is the highest efficiency reported for thin silicon solar cells with a thickness of < 34 μm" such description since we haven't certified it".

I strongly suggest the authors give the certificated report for the reasons: (1) the cells will be applied without the frame, it should be measured at the final application form without frame; (2) the values measured with frame are not correct because of the contribution and influences from the surrounding; (3) the cells can be easily measured at a third-party if they are cut off from the frame and the holder is supplied together with the cells.

Reply: We prepared the TSRR solar cell samples from the very beginning again, and get them certificated with frame since it is easy to break without frame. As for its final application form, it depends on the specific application scenario. If the bending performance requirements are very high, it must be cut off from the frame. If only lightweight is required, it can be applied with or without the frame.

This time we optimized the fabrication process and prepared thinner TSRR solar cells than last time, and realized an efficiency of 20.33% (certified 20.05%) on 28-μm silicon solar cell with all dopant-free and interdigitated back contacts. It is the highest efficiency reported for thin silicon solar cells with a thickness of < 35 μm according to Table 1 in the manuscript. The certification report is shown below.

中国认可
国际互认
检测
TESTING
CNAS L8490

Test and Calibration Center of New Energy Device and Module,
Shanghai Institute of Microsystem and Information Technology,
Chinese Academy of Sciences (SIMIT)

Measurement Report

Report No. 24TR040701

Client Name Shanghai Jiao Tong University

Client Address 800 Dong Chuan Road, Shanghai, China

Sample Thin crystalline silicon solar cell

Manufacturer Shanghai Jiao Tong University, Sun Yat-sen University

Measurement Date 7th April, 2024

Performed by: Qiang Shi *Qiang Shi* **Date:** 07/04/2024

Reviewed by: Wenjie Zhao *Wenjie Zhao* **Date:** 07/02/2024

Approved by: Yucheng Liu *Yucheng Liu* **Date:** 07/04/2024

Address: No.235 Chengbei Road, Jiading, Shanghai **Post Code:**201800

E-mail: solarcell@mail.sim.ac.cn **Tel:** +86-021-69976905

The measurement report without signature and seal are not valid.
This report shall not be reproduced, except in full, without the approval of SIMIT.

**Sample Information**

Sample Type	Thin crystalline silicon solar cell, IBC
Serial No.	05-2-7
Lab Internal No.	24040701-1#
Measurement Item	I-V characteristic
Measurement Environment	24.4 ± 2.0°C, 40.3 ± 5.0%R.H

Measurement of I-V characteristic

Reference cell	PVM1121
Reference cell Type	mono-Si, WPVS, calibrated by NREL (Certificate No. ISO 2098)
Calibration Value/Date of Calibration for Reference cell	143.95mA/ Feb. 2024
Measurement Conditions	Standard Test Condition (STC): Spectral Distribution: AM1.5 according to IEC 60904-3 Ed.3, Irradiance: 1000 ± 50W/m ² , Temperature: 25 ± 2°C
Measurement Equipment/ Date of Calibration	AAA Steady State Solar Simulator (YSS-T155-2M) / July.2023 IV test system (ADCMT 6246) / June. 2023 Measuring Microscope (MF-B2017C) / July.2023 SR Measurement system (CEP-25ML-CAS) / April.2023
Measurement Method	I-V Measurement: Logarithmic sweep in reverse direction (Voc to Isc) during one flash based on IEC 60904-1:2020; Spectral Mismatch factor was calculated according to IEC 60904-7 and I-V correction according to IEC 60891.
Measurement Uncertainty	Area: 1.0%(k=2); Isc: 2.0%(k=2); Voc: 1.0%(k=2); Pmax: 2.4%(k=2); Eff: 2.5%(k=2)

====Measurement Results====

Area [cm ²]	Isc [mA]	Voc [V]	Pmax [mW]	FF [%]	Eff [%]
1.0070	36.517	0.7577	20.186	72.96	20.05

- Spectral Mismatch Factor SMM=0.9958.
- Aperture area defined by a thin black mask was measured by a measuring microscope.
- Test results listed in this measurement report refer exclusively to the mentioned measured samples.
- The results apply only at the time of the test, and do not imply future performance.

Fig.1 I-V curve of the measured sample

-----End of Report-----

We've replaced the old performance data of the TSRR solar cell with the new data in the manuscript, please see changes in Figure 3 of the revised manuscript:

Fig. 3b gives a SEM image of the cross section of the champion thin silicon solar cell in our experiments, its thickness is 28 μm and the height of the random pyramids is ~ 2 μm. The light current density-voltage (J - V) curves of the champion thin cell and the 150-μm control sample are presented in Fig. 3c. A J_{sc} of 36.64 mA · cm⁻², a V_{oc} of 749.1 mV, an FF of 74.08% and an efficiency of 20.33% was achieved on 28-μm silicon solar cell. It is the highest efficiency reported for thin silicon solar cells with a thickness of < 35 μm according to Table 1. The efficiency of the 150-μm control sample was 21.65% with a J_{sc} of 40.37 mA · cm⁻², a V_{oc} of 727.0 mV and an FF of 73.77%. The active areas of both of them are 1.007 cm² due to device limitations in our lab. Despite being more than four times thinner, the champion thin cell achieves 93.9% of the efficiency of the original 150-μm thick silicon control sample. The main gain is a 22.1 mV boost in V_{oc} , which is attributed to a reduction in bulk recombination⁵⁶. The 28-μm thin device achieved a certified efficiency of 20.05% at a credible third-party photovoltaic laboratory (Supplementary Fig. 6). The corresponding reflection (R), external quantum efficiency (EQE), and internal quantum efficiency (IQE) obtained for the champion thin sample are shown in Fig. 3d, and the inset shows the photo of our real solar cell with TSRR structure. The integrated current density J_{EQE} extracted from the EQE is 36.63 mA · cm⁻² and the total reflection is 6.38%. More details and improvement of this type of dopant-free IBC solar cell can be found in our previous works^{55, 57}.

2.I suggest the authors remove “high-success’ in the title, because this technology has not been tested by the industry.

Reply: Thanks for this suggestion, we renamed our title as “Free-standing ultrathin silicon wafers and solar cells through edges reinforcement”.

Reviewer #2:

After the carefully revision, I recommend this manuscript to the Nature Communications.

Reply: We thank the reviewer again for his/her careful review and constructive suggestions for our manuscript, and we are happy to have this positive recommendation.

Reviewer #3:

Overall, I am happy with the answers provided by the authors on the issues raised in the first review, particularly related to the additional experimental work performed on square wafers and industry-relevant processes.

Reply: We thank the reviewer again for his/her careful review and constructive suggestions for our manuscript, and we are happy to have this positive recommendation.

However, I am not yet happy with how prior art is addressed. The work from 1366 should also be cited in the manuscript instead of only in the rebuttal letter. In addition, this paper, Thin HI-ETA/sup (R)/ space silicon solar cells with improved end-of-life performance | IEEE Conference Publication | IEEE Xplore, should be discussed. I could not find the reference at the time of the first review, but I referred to it as work in the space silicon area. I will leave it up to the editor to assess how this affects the novelty of this work.

Reply: We apologize for not fully understanding of your suggestions at the time of the first review. Prior similar works should be discussed in the manuscript as you suggested, please see changes in introduction sector of the revised manuscript:

For example, a locally thinned waffle-like cell was proposed for space silicon solar cell in 2000. Strobl *et al.* reported a 15.8% efficiency silicon solar cell with a thickness of 50 μm in the locally thinned regions and 130 μm for the frames¹. But other details of this structure are particularly underreported. There is also a “3-D” wafer technology developed by 1366 technology, Inc. around 2016. It is a multi-crystalline silicon wafer growing technology which forms a wafer directly from molten silicon in a bath-like furnace, with the ability to locally control wafer thickness. Thus, it can produce thin

wafers with thick edge^{2,3}. However, it suffers from serious problems: low bulk lifetime, high total thickness variation (TTV) and difficulty in growing very thin framed wafers³.

In addition, there are several typos/grammar mistakes in the revised version of the manuscript (e.g., rete instead of rate) that should be addressed.

Reply: We thank the reviewer very much for his/her careful review of the manuscript. We checked the article carefully again and those mistakes have been addressed.

References:

1. Strobl, G. et al. Thin HI-ETA space silicon solar cells with improved end-of-life performance. *28th IEEE Photovolt. Spec. Conf.* 1289-1292 (2000).
2. Lorenz, A., Hofstetter, J., Malkasian, H., Sanderson & L., Mierlo, V. F. 3 Dimensional Direct Wafer product with locally-controlled thickness. *32nd Europe. Photovolt. Sol. Energ. Conf. Exhibi.* 310-312 (2016).
3. Lorenz, A. 1366 Project Automate: Enabling Automation for < \$0.10/W High-Efficiency Kerfless Wafers Manufactured in the US. 1366 Technologies, Bedford, MA, United States (2017).